# Mixed-Features Vectors & Subspace Splitting

**Alejandro Pimentel-Alarcón**
IIMAS, Universidad Nacional
Autónoma de México, UNAM
Mexico City, Mexico
pimentel@comunidad.unam.mx

**Daniel Pimentel-Alarcón**
Department of Biostatistics
Wisconsin Institute for Discovery
UW-Madison, WI, 53715
pimentelalar@wisc.edu

## Abstract

Motivated by metagenomics, recommender systems, dictionary learning, and related problems, this paper introduces *subspace splitting* (Ss): the task of clustering the entries of what we call a *mixed-features vector*, that is, a vector whose subsets of coordinates agree with a collection of subspaces. We derive precise identifiability conditions under which Ss is well-posed, thus providing the first fundamental theory for this problem. We also propose the first three practical Ss algorithms, each with advantages and disadvantages: a random sampling method , a projection-based greedy heuristic , and an alternating Lloyd-type algorithm ; all allow noise, outliers, and missing data. Our extensive experiments outline the performance of our algorithms, and in lack of other Ss algorithms, for reference we compare against methods for tightly related problems, like robust matched subspace detection and maximum feasible subsystem, which are special simpler cases of Ss.

## 1 Introduction

As the reach of data science expands, and as we continuously improve our sensing, storage and computing capabilities, data in virtually all fields of science keeps becoming increasingly high-dimensional. For example, the CERN Large Hadron Collider currently "generates so much data that scientists must discard the overwhelming majority of it, hoping that they've not thrown away anything useful" [1], and the upcoming Square Kilometer Array is expected to produce 100 times that [2]. Fortunately, high-dimensional data often has an underlying low-dimensional structure. Inferring such structure not only cuts memory and computational burdens, but also reduces noise and improves learning and prediction. However, higher dimensionality not only increases computational requirements; it also augments data's structure complexity. In light of this, several research lines have explored new low-dimensional models that best summarize data, going from principal component analysis (PCA) [3–11] and single subspaces [12–21] to unions of subspaces [22–40], and algebraic varieties [41].

This paper introduces *mixed-features vectors* (Mfv's): a new model that describes the underlying structure of data arising from several modern applications that is not captured by existing low-dimensional models. The main idea is that each entry of a Mfv comes from one out of several classes, and that the entries of the same class lie in an underlying subspace. In particular, Mfv's are motivated by megatenomics [42–46] and recommender systems [47–59]: in metagenomics each gene segment comes from one of the several taxa present in a microbiome; in recommender systems each rating may come from one of several users sharing the same account. However, Mfv's also have applications in robust estimation (e.g., robust PCA [3–11] and robust dictionary learning [60–67]), matrix completion [48–59], subspace clustering [22–39], and more.

This paper also introduces *subspace splitting* (Ss): the task of clustering the entries of a Mfv according to its underlying subspaces. Ss is tightly related to other machine learning problems. In particular, Ss can be thought as a generalization of robust matched subspace detection (Rmsd) [12–17], and maximum feasible subsystem (MaxFS) [68–74]. However, the added complexity of Ss renders existing approaches for these problems inapplicable, which calls the attention for specialized Ss theory and methods. In these regards, (i) we derive precise identifiability conditions under which Ss is well-posed, and (ii) we propose the first three Ss algorithms.

## 2   PROBLEM STATEMENT AND FUNDAMENTAL THEORY

Let $\mathbb{U}^1, \ldots, \mathbb{U}^K$ be subspaces of $\mathbb{R}^d$, and let $\Omega_0, \Omega_1, \ldots, \Omega_K$ denote a partition of $[d] := \{1, \ldots, d\}$. For any subspace, matrix or vector that is compatible with a set of indices $\Omega \subset [d]$, we will use the subscript $\Omega$ to denote its restriction to the coordinates/rows in $\Omega$. For example, $\mathbb{U}^1_{\Omega_K} \subset \mathbb{R}^{|\Omega_K|}$ denotes the restriction of $\mathbb{U}^1$ to the coordinates in $\Omega_K$. Define $\mathbf{x} \in \mathbb{R}^d$ as the *mixed-features vector* (MFV) such that $\mathbf{x}_{\Omega_k} \in \mathbb{U}^k_{\Omega_k}$ for each $k = 1, \ldots, K$, and the entries of $\mathbf{x}_{\Omega_0}$ are outliers. Let $\boldsymbol{\epsilon} \in \mathbb{R}^d$ denote a noise vector with variance $\sigma^2$. Given $\mathbb{U}^1, \ldots, \mathbb{U}^K$, and an incomplete observation $\mathbf{y}_\Omega = \mathbf{x}_\Omega + \boldsymbol{\epsilon}_\Omega$, the goal of *subspace splitting* (SS) is to determine the subsets $\Omega_1 \cap \Omega, \ldots, \Omega_K \cap \Omega$ indicating the observed coordinates of $\mathbf{y}$ that match with each subspace.

**Example 1.** *Consider the following setup, with $1$-dimensional subspaces $\mathbb{U}^1, \mathbb{U}^2$ spanned by $\mathbf{U}^1, \mathbf{U}^2$:*

$$
\mathbf{U}^1 = \begin{bmatrix} 1 \\ 1 \\ 1 \\ 1 \\ 1 \\ 1 \end{bmatrix}, \quad
\mathbf{U}^2 = \begin{bmatrix} 1 \\ 2 \\ 3 \\ 4 \\ 5 \\ 6 \end{bmatrix}, \quad
\mathbf{x} = \begin{bmatrix} 1/2 \\ 1/2 \\ 6 \\ 8 \\ 9 \\ 10 \end{bmatrix}, \quad
\boldsymbol{\epsilon} = \begin{bmatrix} 0.1 \\ -0.1 \\ -0.1 \\ 0.1 \\ -0.1 \\ 0.1 \end{bmatrix}, \quad
\mathbf{y}_\Omega = \begin{bmatrix} 0.51 \\ 0.49 \\ 5.9 \\ 8.1 \\ 8.9 \\ \cdot \end{bmatrix}.
$$

*It is easy to see that $\Omega_1 = \{1, 2\}$, $\Omega_2 = \{3, 4\}$, $\Omega_0 = \{5\}$, because $\mathbf{x}_{\Omega_1} = \frac{1}{2}\mathbf{U}^1_{\Omega_1}$ and $\mathbf{x}_{\Omega_2} = 2\mathbf{U}^2_{\Omega_2}$.*

The keen reader will immediately wonder: is there another partition $\{\Omega'_1, \ldots, \Omega'_K\}$ different from $\{\Omega_1, \ldots, \Omega_K\}$ such that $\mathbf{x}_{\Omega'_k} \in \mathbb{U}^k_{\Omega'_k}$ for every $k$? In other words, is this problem well-posed, and if so, under what conditions? Our main theoretical result answers this question, showing that under the next assumptions, $\Omega_k$ can be recovered if and only if it has more elements than the dimension of $\mathbb{U}^k$.

**A1** Each $\mathbb{U}^k$ is drawn independently with respect to the uniform measure over the Grassmannian.

**A2** Each $\mathbf{x}_{\Omega_k}$ is drawn independently according to an absolutely continuous distribution with respect to the Lebesgue measure on $\mathbb{U}^k_{\Omega_k}$.

In words, **A1** essentially requires that $\mathbb{U}^1, \ldots, \mathbb{U}^K$ are in general position with no particular relation with one another. Similarly, **A2** requires that each *piece* of $\mathbf{x}$ is in general position over its corresponding *piece* of subspace. This type of *genericity* assumptions are becoming increasingly common in compressed sensing, matrix completion, subspace clustering, tensor theory, and related problems [10, 21, 30, 31, 33–38, 41, 56–59]. All our statements hold with probability $1$ with respect to the measures in **A1** and **A2**. We point out that **A1** and **A2** do not imply coherence or affinity (other typical assumptions in related theory that quantify alignment with the canonical axes or between subspaces [3, 6, 11, 23, 26, 27, 48–50, 54, 55]) nor vice-versa. For example, bounded coherence and affinity assumptions indeed allow subspaces perfectly aligned on some coordinates. However, they rule-out cases that our assumptions allow, for example the non-zero measure set of highly coherent or affine subspaces that are *somewhat* aligned with the canonical axes or with one another. To sum up, these assumptions are different, not stronger nor weaker than the usual coherence and affinity assumptions. With this, we are ready to state our main theorem, showing that subspace splitting is possible if and only if $\mathbf{x}$ contains more than $\dim(\mathbb{U}^k)$ entries of each subspace $\mathbb{U}^k$.

**Theorem 1.** *Suppose A1 and A2 hold. Given $\mathbf{x}$ and $\mathbb{U}^1, \ldots, \mathbb{U}^K$, one can identify $\Omega_1, \ldots, \Omega_K$ if and only if $|\Omega_k| > \dim(\mathbb{U}^k)$ for every $k$.*

Example 1 shows a case where the conditions of Theorem 1 are met ($|\Omega_1| = |\Omega_2| = 2 > \dim(\mathbb{U}^1) = \dim(\mathbb{U}^2) = 1$), and consequently subspace splitting is well-posed (there exist no partition other than the true $\{\Omega_1, \Omega_2\}$ that splits $\mathbf{x}$ into $\mathbb{U}^1$ and $\mathbb{U}^2$). Conversely, the following Example shows a case where the conditions of Theorem 1 are not satisfied, and at least some $\Omega_k$ is unidentifiable.

**Example 2.** *Consider the following setting:*

$$\mathbf{U}^1 = \begin{bmatrix} 1 \\ 1 \\ 1 \\ 1 \end{bmatrix}, \quad \mathbf{U}^2 = \begin{bmatrix} 1 \\ 2 \\ 3 \\ 4 \end{bmatrix}, \quad \mathbf{U}^3 = \begin{bmatrix} 0 \\ 1 \\ 4 \\ 9 \end{bmatrix}, \quad \mathbf{x} = \begin{bmatrix} 1/2 \\ 1/2 \\ 6 \\ 3 \end{bmatrix}.$$

*Now $|\Omega_2| = |\Omega_3| = 1 = \dim(\mathbb{U}^2) = \dim(\mathbb{U}^3)$. As Theorem 1 shows, there exist multiple ways to split $\mathbf{x}$ into $\mathbb{U}^1$, $\mathbb{U}^2$, and $\mathbb{U}^3$. Here the partitions could be $\{\Omega_1, \Omega_2, \Omega_3\} = \{\{1,2\}, \{3\}, \{4\}\}$ or $\{\{1,2\}, \{4\}, \{3\}\}$, and there is no way of telling which is the* true *partition from $\mathbb{U}^1$, $\mathbb{U}^2$, $\mathbb{U}^3$, and $\mathbf{x}$. In other words, $\Omega_2$ and $\Omega_3$ are unidentifiable.*

**Remark 1.** *We point out that the constructions in our examples are not generic (i.e., they were not constructed according to **A1** and **A2**). We chose them for their simplicity to build intuition and make a point. However, our results still apply to these constructions, showing that in addition to* all *generic cases, our theory also holds for* some *non-generic ones. An exact characterization of* all *the non-generic cases that our theory covers requires a careful study of notions of sketching and partial coordinate discrepancy [31], which are out of the scope of this paper.*

The proof of Theorem 1 follows by the next two lemmas. Lemma 1 shows that *any* subset of $r$ or fewer entries of *any* vector will always match with *any* $r$-dimensional subspace in general position. Lemma 2 shows that $r + 1$ entries of a vector won't match with a random $r$-dimensional subspace by chance. Said in other words, $r + 1$ entries of a vector will match with an $r$-dimensional subspace if and only if such entries truly come from that $r$-dimensional subspace. Lemma 2 is effectively the key towards Theorem 1, as it allows us to try all combinations of $r + 1$ entries that fit in an $r$-dimensional subspace, knowing that we will never get a false match. All proofs are in Appendix A.

**Lemma 1.** *Suppose **A1** holds. Let $\Omega \subset [d]$. If $|\Omega| \leq \dim(\mathbb{U}^k)$, then $\mathbf{x}_\Omega \in \mathbb{U}_\Omega^k$ for every $\mathbf{x} \in \mathbb{R}^d$.*

**Lemma 2.** *Suppose **A1** and **A2** hold. Let $\Omega$ be an arbitrary subset of $[d]$ with exactly $\dim(\mathbb{U}^k) + 1$ elements. Then $\mathbf{x}_\Omega \in \mathbb{U}_\Omega^k$ if and only if $\Omega \subset \Omega_k$.*

Corollary 1 extends these results to account for noise and missing data, replacing **A1** and **A2** with:

**A1'** The coherence of $\mathbb{U}^k$ is upper bounded by $\mu$, and the geodesic distance over the Grassmannian between $\mathbb{U}^k$ and $\mathbb{U}^\ell$ is lower bounded by $\varphi$, for every $k, \ell = 1, \ldots, K$.

**A2'** The coherence of $\mathbf{x}_{\Omega_k}$ is upper bounded by $\nu$, and its norm is lower bounded by $\psi$, for every $k = 1, \ldots, K$.

**Corollary 1.** *Suppose **A1'** and **A2'** hold with $\mu, \nu < C\sigma$, and $\varphi, \psi > c\sigma$ for some constants $C$ and $c$. Let $\mathbf{y}_\Omega$ and $\mathbb{U}^1, \ldots, \mathbb{U}^K$ be given. Suppose $|\Omega_k \cap \Omega| > \dim(\mathbb{U}^k)$ for every $k$. Then with probability decreasing in $C$ and increasing in $c$, one can identify $\Omega_1 \cap \Omega, \ldots, \Omega_K \cap \Omega$.*

To guarantee identifiability, Corollary 1 requires that subspaces and samples are sufficiently incoherent and separated to overcome the noise level $\sigma$. Notice that since now there is missing data (we observe $\mathbf{y}_\Omega$, instead of $\mathbf{x}$ as in Theorem 1), Corollary 1 requires that there are enough *observed* entries per subspace, i.e., that $|\Omega_k \cap \Omega|$ are sufficiently large (rather than $\Omega_k$). Similarly, only the observed entires can be classified, so only the intersections $\Omega_k \cap \Omega$ are identifiable (rather than $\Omega_k$).

## 3 RELATED WORK

Arguably, the closest link that subspace splitting has with other popular machine learning problems, is through robust matched subspace detection (RMSD) [12–17] and maximal feasible subsystem (MAXFS) [68–74]. In RMSD one observes a vector $\mathbf{x} \in \mathbb{R}^d$ of the form $\mathbf{x} = \mathbf{U}\boldsymbol{\theta} + \mathbf{z}$, where $\mathbf{z} \in \mathbb{R}^d$ is a sparse vector of *outlier* entries, and $\boldsymbol{\theta} \in \mathbb{R}^r$ is the coefficient of the entries of $\mathbf{x}$ that match with the subspace spanned by $\mathbf{U} \in \mathbb{R}^{d \times r}$. Given $\mathbf{x}$ and $\mathbf{U}$, the goal is to equivalently find $\boldsymbol{\theta}$, $\mathbf{z}$, or the support of $\mathbf{z}$, which in turn determine the coordinates $\Omega$ where $\mathbf{x}$ matches $\mathbf{U}$. Subspace splitting can be thought as the generalization of RMSD to the case where there are multiple subspaces, and we want to find the entries that match each one. Similarly, in MAXFS one has an inconsistent system of equations of the form $\mathbf{x} = \mathbf{U}\boldsymbol{\theta}$, and wants to determine the solution $\boldsymbol{\theta}$ that produces the largest

subset $\Omega$ of consistent equations. Subspace splitting can be thought as the generalization of MAXFS to the case where there are multiple consistent subsystems (one per subspace), and we want to find each of them. The three prevalent venues to solve these problems can be classified into three groups: (i) $\ell_1$ minimization, (ii) mixed integer programs, and (iii) random sampling.

The first group essentially encompasses variants of minimizing $\|\mathbf{x} - \mathbf{U}\boldsymbol{\theta}\|_1$ over $\boldsymbol{\theta} \in \mathbb{R}^r$, which is tightly related to the Lasso [91]. The intuition is that the $\ell_1$-norm will favor solutions that produce a sparse vector $\mathbf{z} = \mathbf{x} - \mathbf{U}\boldsymbol{\theta}$ whose zero entries reveal $\Omega$. While formulations like this work great for just one subspace (or subsystem), if there are two or more, then for each subspace, the entries of all other subspaces are outliers, in which case $\mathbf{z}^k := \mathbf{x} - \mathbf{U}^k\boldsymbol{\theta}^k$ will no longer be sparse, and the solution to the $\ell_1$-norm minimization will no longer reveal $\Omega_k$. The second group aims to directly recover $\Omega$ with variants of the mixed integer formulation that minimizes $\|\mathbf{z}\|$ subject to $|\mathbf{x} - \mathbf{U}\boldsymbol{\theta}| \leq \mathrm{M}\mathbf{z}$ over $\mathbf{z} \in \{0, 1\}^d$ and $\boldsymbol{\theta} \in \mathbb{R}^r$, where $\mathrm{M}$ is a tuning parameter. Here the main idea is to maximize the number of zero entries in $\mathbf{z}$ (which reveal $\Omega$), where we are forcing $\mathbf{x}$ to match $\mathrm{span}\{\mathbf{U}\}$, because the constraint guarantees that $|\mathbf{x}_i - \mathbf{U}\boldsymbol{\theta}| \leq \mathbf{z}_i = 0$. Unfortunately, because of the combinatorial nature of this approach, this method becomes exponentially slow as the fraction of outliers/inconsistencies increases, rendering it prohibitive in practice for even small values of K. The third group essentially uses the same principle as random sampling consensus (RANSAC) [92–98]. That is, one iteratively selects a candidate subset of entries at random, until it finds one where $\mathbf{x}$ agrees with $\mathrm{span}\{\mathbf{U}\}$. This is in fact the approach that we used in the proof of Theorem 1, and the main idea behind our first SS algorithm, which we present in Section 5.1, and study in Section 6.

**Remark 2.** *We point out that SS is also related to subspace clustering [22–40]. The main difference is that in subspace clustering all entries of each column come from the same subspace.*

## 4 MOTIVATING APPLICATIONS

This section details the main motivations behind SS, namely, metagenomics, recommender systems, and robust estimation, relevant for dictionary learning, matrix completion, and related problems.

**Metagenomics** Microbial communities affect the balance of entire ecosystems, and play a crucial role in the planet's health [42]. Consequently, understanding microbiome compositions, interactions, and evolutions holds the key to the knowledge of the deep interconnectivity factors that play a role on Earth's biodiversity, our agricultural sustainability, and adaptation to global threats like climate change. One cornerstone tool towards microbiome understanding lies in genomic analysis. In practice, to obtain an organism's genome (e.g., a person's genome), biologists feed a DNA sample (e.g., blood or hair) to a sequencer machine that produces a series of *reads*, which are short genomic sequences that can later be assembled and aligned to recover the entire genome. The challenge arises when the sequencer is provided a sample with DNA from multiple organisms, as is the case in any microbiome (e.g., the human gut microbiome), where any sample will contain a mixture of DNA from multiple taxa that cannot be trivially classified [43]. In this case, each read produced by the sequencer may correspond to a different taxa, resulting in a DNA sample with a mixture of genes. Existing approaches depend on the correct classification of taxonomic units, which in turn relies on the existence of reference tables, and often require human intervention [44]. However, reference sequences remain unknown for a vast majority of microbial biodiversity, which in turn precludes holistic metagenomic analyses, as scientists often have to discard unidentified data that cannot be currently categorized with existing reference tables [45]. In contrast, our work pioneers the theoretical foundations of a novel model tailored to automatically classify taxonomic units without any human intervention. Moreover, our model will naturally allow for missing data, and hence it will be robust to taxa with length-varying sequences, and fast-occurring mutations, which are in fact quite common in several microorganisms such as certain types of bacteria [46].

**Recommender Systems and Image Inpainting** In recommender systems like Amazon or Netflix, one aims to infer users' preferences in order to make good recommendations [47]. Arguably the most renowned model for this task is low-rank matrix completion [48–58], which assumes that each user's preferences vector can be explained as a linear combination of a few others. Said in other words, preferences lie in a linear subspace. However, often multiple types of users share the same account. In this case the vector of preferences will contain a mixture of entries from multiple subspaces. It is easy to see (details in the proof of Theorem 1 in Appendix A) that the coefficient corresponding to the entries in the $k^{th}$ subspace are given by $\boldsymbol{\theta}^k = (\mathbf{U}_{\Omega'_k}^k)^{-1}\mathbf{x}_{\Omega'_k}$, where $\Omega'_k$ is any subset of $\Omega_k$ with

more than $\dim(\mathbb{U}^k)$ elements. This insight can be used to estimate the values corresponding to the $k^{th}$ subspace that *would* appear in the locations where there are observations from other subspaces, or where data is missing. In For example, in recommender systems, this would allow to infer the preferences of *all* users sharing an account. In image inpainting this would allow to estimate the missing, corrupted, or occluded pixels in an image [60–67].

**Robust Learning** In Section 3 we showed that Ss is a generalization of RMSD and MAXFS, both of which are crucial subtasks in many importantproblems, including robust PCA [3–11], dictionary learning [60–67], matrix completion [48–58], and more. For example, in subspace tracking one needs to identify outliers in each new vector (meaning performing RMSD or MAXFS) before updating the underlying subspace [18–20]. Or in subspace clustering one needs to identify outliers in each vector before finding sparse representations [26, 28, 29, 32, 39]. These in turn are core techniques in target localization [76], medical imaging [79], communications [80], anomaly detection [81, 82], hyperspectral imaging [16], and networks estimation [84? –89], among many others [78]. Being a generalization of RMSD and MAXFS, subspace splitting is also applicable in these broader problems, bringing possibilities for improved performance.Moreover, as these problems evolve into more sophisticated models that allow mixed-features vectors (such as mixture matrix completion [59]), subspace splitting will gain importance and become a crucial subroutine of these emerging problems.

## 5 SUBSPACE SPLITTING ALGORITHMS

We now present our three subspace splitting algorithms: a random sampling method, an iterative projection-based greedy heuristic, and an alternating approach inspired by Lloyd's algorithm [75].

### 5.1 RANDOM SAMPLING SPLITTING

Like Theorem 1 suggests, for each $k$ we can iteratively try sets $\Omega'_k$ with $\dim(\mathbb{U}^k) + 1$ entries selected randomly until we find one such set where $\mathbf{y}_{\Omega'_k}$ lies close to $\mathbb{U}^k_{\Omega'_k}$ (within the noise level $\sigma$). At that point we can estimate $\boldsymbol{\theta}^k = (\mathbf{U}^k_{\Omega'})^{-1}\mathbf{y}_{\Omega'}$, and subsequently $\Omega_k$ as the entries in $\mathbf{y}_\Omega - \mathbf{U}^k_\Omega \boldsymbol{\theta}^k$ that are within $\sigma$. We call this approach *random sampling splitting* (RANSAS). The main advantage of RANSAS is that as consequence of Theorem 1, it is *guaranteed* to work after enough iterations. To see this, observe that if entries in $\mathbf{x}$ are uniformly distributed across subspaces and outliers, then the probability of randomly selecting $r + 1$ entries from the $k^{th}$ subspace is $1/(K+1)^{r+1}$. Consequently, the probability that $r + 1$ random entries come from the same subspace is $K/(K+1)^{r+1}$. A simple Chernoff bound shows that the expected number of iterations before this happens is $\mathcal{O}(K^r)$. Since we want this to happen $K$ times, we conclude that the expected number of iterations is upper bounded by $\mathcal{O}(K^{r+1})$. We summarize this in the next theorem.

**Theorem 2.** *Let A1 and A2 hold. Suppose* $|\Omega_k| > \dim(\mathbb{U}^k)$ *for every* $k$. *Suppose* $\mathbb{P}(i \in \Omega_k) = 1/(K+1)$ *independently for every* $i, k$. *Then the expected number of iterations before* RANSAS *identifies* $\Omega_1, \ldots, \Omega_K$ *is lower and upper bounded by* $\mathcal{O}(K^{\min_k \dim(\mathbb{U}^k)+1})$ *and* $\mathcal{O}(K^{\max_k \dim(\mathbb{U}^k)+1})$.

We point out that a scenario where all entries are equally distributed is in fact the most difficult setting. Otherwise, identifying any subspace with probability $p_k > 1/(K+1)$ will take $\mathcal{O}(1/p_k^{r+1}) < \mathcal{O}(1/(K+1)^{r+1})$ iterations to find. The main downside of this approach is that its computational complexity scales exponentially. So if $K$ and $r$ are small, or there is one dominant subspace, RANSAS may be a good idea. However, as we see in our experiments, this approach quickly becomes computationally prohibitive as $K$ and $r$ grow.

### 5.2 GREEDY SPLITTING

The main idea of our *greedy splitting* (GREEDYS) heuristic is to iteratively remove the most disruptive entry from each subspace. More precisely, for each $k$ we will start with $\Omega^0_k = [d]$, and for each $t > 0$ we will define $\Omega^t_k := \Omega^{t-1}_k \backslash i^{t-1}$, where $i^{t-1}$ is the entry whose removal from $\Omega^{t-1}_k$ minimizes the difference between $\mathbf{y}_{\Omega^t_k}$ and its projection onto $\mathbb{U}^k_{\Omega^t_k}$. The intuition is that removing $i^{t-1}$ gets $\mathbf{y}_{\Omega^t_k}$ closer to $\mathbb{U}^k_{\Omega^t_k}$. We repeat this procedure for each $k$ until $\mathbf{y}_{\Omega^t_k}$ is close to $\mathbb{U}^k_{\Omega^t_k}$ (within the noise level $\sigma$). We know from Lemma 1 that this procedure will terminate at some point, because *any* subset of $r$

or fewer entries of *any* vector will always match with *any* r-dimensional subspace in general position. If for some k this procedure terminates with a set $\Omega_k^t$ with more than $r_k = \dim(\mathbb{U}^k)$ entries, Theorem 1 suggests that $\mathbf{y}_{\Omega_k^t}$ truly lies in $\mathbb{U}_{\Omega_k^t}^k$, that $\Omega_k^t \subset \Omega$, that we can estimate $\boldsymbol{\theta}^k = (\mathbf{U}_{\Omega_k^t}^k)^{-1}\mathbf{y}_{\Omega_k^t}$, and identify $\Omega_k$ as the entries in $\mathbf{y}_\Omega - \mathbf{U}_\Omega^k\boldsymbol{\theta}^k$ that are within $\sigma$. At this point we can prune all the entries corresponding to this subspace, and repeat the greedy procedure from the start without these entries. If at some point none of the remaining sets $\Omega_k^t$ has more than $r_k$ entries, we know by Lemma 1 that we cannot determine whether $\mathbf{y}_{\Omega_k^t}$ truly lies in $\mathbb{U}_{\Omega_k^t}^k$ with these entries alone, then we stop and consider this a failure. Notice that for each k, GREEDYS removes the most disruptive entry among all d entries, then the next most disruptive among the remaining $d-1$, until either all the remaining entries lie in the same subspace, or there remain no more than $r_k$ entries. This requires no more than $d+(d-1)+(d-2)+\cdots+(r_k+1) = \mathcal{O}(d^2)$ iterations. After identifying the entries of a subspace, if pruning is necessary, we potentially need to repeat the iterative removal process for the remaining $K-1$ subspaces, and so on. In total, we potentially need to repeat the iterative removal process $K+(K-1)+(K-2)+\cdots+1 = \mathcal{O}(K^2)$ times. Putting these two observations together, we obtain the following theorem, showing that GREEDYS has a polynomial computational complexity.

**Theorem 3.** GREEDYS *will terminate after no more than $\mathcal{O}(Kd)^2$ iterations.*

The main caveat of this greedy approach, as the next Example shows, is that even in a noiseless setting, the most disruptive entry for $\mathbb{U}^k$ (that is, the entry whose removal minimizes the distance between $\mathbf{y}_{\Omega_k^t}$ and its projection onto $\mathbb{U}_{\Omega_k^t}^k$) may in fact correspond to $\mathbb{U}^k$.

**Example 3.** *Consider the following construction where the first two entries of $\mathbf{x}$ correspond to $\mathbf{U}^1$, the last two correspond to $\mathbf{U}^2$, there are no outliers, i.e., $\Omega_0 = \emptyset$, there is no noise, i.e., $\boldsymbol{\epsilon} = \mathbf{0}$, and there is no missing data, i.e., $\Omega = [d]$, so that we observe all entries of $\mathbf{y} = \mathbf{x}$:*

$$\mathbf{U}^1 = \begin{bmatrix} 2 \\ 2 \\ 3 \\ 4 \end{bmatrix}, \qquad \mathbf{U}^2 = \begin{bmatrix} 1 \\ 3 \\ 1 \\ 1 \end{bmatrix}, \qquad \mathbf{y} = \begin{bmatrix} 4 \\ 4 \\ 2 \\ 2 \end{bmatrix}.$$

*Pay special attention to $\mathbf{U}^1$ and $\mathbf{y}$. Let $\Omega_{-i} := [d]\backslash i$ denote the set of all except the $i^{th}$ entry, and let $\hat{\mathbf{y}}_{\Omega_{-i}}^k := \mathbf{U}_{\Omega_{-i}}^k(\mathbf{U}_{\Omega_{-i}}^{k\mathsf{T}}\mathbf{U}_{\Omega_{-i}}^k)^{-1}\mathbf{U}_{\Omega_{-i}}^{k\mathsf{T}}\hat{\mathbf{y}}_{\Omega_{-i}}$ denote the projection of $\mathbf{y}_{\Omega_{-i}}$ onto $\mathbb{U}_{\Omega_{-i}}^k$. For example, here:*

$$\mathbf{U}_{\Omega_{-1}}^1 = \begin{bmatrix} 2 \\ 3 \\ 4 \end{bmatrix}, \qquad \mathbf{y}_{\Omega_{-1}} = \begin{bmatrix} 4 \\ 2 \\ 2 \end{bmatrix}, \qquad \hat{\mathbf{y}}_{\Omega_{-1}}^1 = \begin{bmatrix} 1.572 \\ 2.2759 \\ 3.0345 \end{bmatrix}.$$

*Proceeding according to our greedy heuristic, we can compute the projection residuals when removing each entry:*

$$\|\mathbf{y}_{\Omega_{-1}} - \hat{\mathbf{x}}_{\Omega_{-1}}^1\| = 2.7038, \qquad \|\mathbf{y}_{\Omega_{-2}} - \hat{\mathbf{x}}_{\Omega_{-2}}^1\| = 2.7038,$$

$$\|\mathbf{y}_{\Omega_{-3}} - \hat{\mathbf{x}}_{\Omega_{-3}}^1\| = 3.4641, \qquad \|\mathbf{y}_{\Omega_{-4}} - \hat{\mathbf{x}}_{\Omega_{-4}}^1\| = 2.7440.$$

*Here we see that removing any of the first two entries makes $\mathbf{y}$ appear closer to $\mathbb{U}^1$, even though the first two entries in fact correspond to $\mathbb{U}^1$. We thus conclude that our greedy heuristic, while it works well in practice, it has no theoretical guarantee of working.*

## 5.3 K-SPLITS

As the name suggests, our last algorithm, K-SPLITS, is inspired by the K-means algorithm. The main idea is to initialize partitioning randomly the entries of $\mathbf{y}_\Omega$ into $\hat{\Omega}_0, \hat{\Omega}_1, \ldots, \hat{\Omega}_K$, and then alternate until convergence between (i) estimating coefficients $\hat{\boldsymbol{\theta}}^1, \ldots, \hat{\boldsymbol{\theta}}^K$ in light of the assignment of entries, and (ii) reassigning entries according to these coefficients. More precisely, at each step after pruning when possible we:

(i) Estimate coefficients $\hat{\boldsymbol{\theta}}^k = (\mathbf{U}_{\hat{\Omega}_k}^k)^{-1}\mathbf{y}_{\hat{\Omega}_k}$.

(ii) Compute $\hat{\mathbf{y}}^k := \mathbf{U}^k\hat{\boldsymbol{\theta}}^k$, and assign entries as $\hat{\Omega}_k = \{i \in [d] : |\mathbf{y}_i - \hat{\mathbf{y}}_i^k| \leq \tau(\sigma)\}$, where $\tau(\sigma)$ is a tuning thresholding parameter that depends on the noise level.

The assignment step is done on a cyclic manner, that is, in every iteration, each cluster is assigned the entry that agrees the most with it, and it receives no new entries until the next iteration. This changes the isotropic assumption of K-means with the assumption that each cluster has the same number of entries, as mentioned in Section 5.1, this is the most difficult setting for this problem, and whenever the assumption is not fulfilled, the algorithm will still be able to complete at least the most dominant subspace even if entries of that subspace contaminate other subspaces; for the next iteration that won't be the case any more. Thanks to this, the assumption of every subspace having the same number of entries does not affect the effectiveness of the algorithm.

As we will see in our experiments, this method outperforms the rest. Its main downside, however, is that like most alternating methods, it suffers from local minima, and depends heavily on initialization. To mitigate the former, in our experiments we run several independent random initializations. We observe that in practice this has excellent performance. We conjecture that it is because this strategy will at some point spread the *centers* $\boldsymbol{\theta}^k$ enough, in light of the subspaces. This sort of spread is crucial in other related clustering algorithms, such as K-means [75]. Our future work will further investigate this, convergence, complexity, and initialization strategies, such as parallels of K-means++ [99], which require careful analysis and are out of the scope of this paper.

# 6 EXPERIMENTS

In this section we study the performance of our three methods above, as a function of the number of subspaces K, the number of entries per subspace $|\Omega_k|$ (which in turn is a proxy of the ambient dimension d) and the subspaces dimensions r. We measure error as the fraction of misclassified entries. In lack of other existing subspace splitting methods, for reference we compare them against the $\ell_1$ minimization described in Section 3. Our results show that our SS methods perform as well as existing methods in the single subspace case, and succeed in the presence of multiple subspaces, where existing methods fail dramatically. In the interest of reproducibility, all our code is available here [100].

In our experiments we first generate matrices $\mathbf{U}^1, \ldots, \mathbf{U}^K \in \mathbb{R}^{d \times r}$ with i.i.d. $\mathcal{N}(0,1)$ entries, to use as bases of $\mathbb{U}^1, \ldots, \mathbb{U}^K$, with $d = (K+1)|\Omega_k|$. Similarly, we generate $\boldsymbol{\theta}^1, \ldots, \boldsymbol{\theta}^K \in \mathbb{R}^r$, also with i.i.d. $\mathcal{N}(0,1)$ entries, to use as coefficient vectors. Next we create the partition of entries $\{\Omega_0, \Omega_1, \ldots, \Omega_K\}$ where each $\Omega_k$ has the exact same number of elements. As discussed in Section 5.1, this is in fact the most difficult setting; otherwise the dominant subspace is easier to find, and one can simply prune its corresponding entries to simplify the problem. Then we create $\mathbf{x}$ as the vector such that $\mathbf{x}_{\Omega_k} = \mathbf{U}_{\Omega_k}^k \boldsymbol{\theta}^k$ for each k, and $\mathbf{x}_{\Omega_0}$ are i.i.d. $\mathcal{N}(0,1)$ entries representing outliers. Finally, we generate $\boldsymbol{\epsilon} \in \mathbb{R}^d$ with i.i.d. $\mathcal{N}(0, \sigma^2)$ entries and $\Omega \subset d$ with a fraction p of observed entries, to construct our observed data $\mathbf{y}_\Omega = \mathbf{x}_\Omega + \boldsymbol{\epsilon}_\Omega$.

In our first experiment we study the behavior of our algorithms as a function of K, with r = 5, $|\Omega_k| = 40$, p = $^1/_2$ (so that on average 20 observed entries correspond to each subspace), and $\sigma = 0$, i.e., no noise. Figure 1 (a) shows that even for K = 2 (the smallest value of K), existing methodology is unable to split $\mathbf{y}$. This is not surprising, as existing methods are meant for settings where *most* of the entries in $\mathbf{y}$ agree with a single subspace, and there are only a few outliers. In our setting, for each subspace, the entries of all other subspaces are outliers. Consequently, a fraction $1 - ^1/_K$ of the entries (at least half with K = 2, and a dominant majority as K increases) are outliers for each subspace, which is well-documented to make existing methods fail (see the discussion in Section 3 for details). In contrast, GREEDYS is better with K = 6 (which amounts to 83.3% outliers) than $\ell_1$ is with K = 2, while K-SPLITS and RANSAS maintain a 100% success rate for every K we tried. Nonetheless, increasing in K comes at a price (time), as seen in Figure 1 (a'), which shows how the number of iterations of each algorithm grows with K, relative to the simplest case (K = 2). Notice that the speed of K-SPLITS decays nicely with K. For instance, with K = 6, K-SPLITS degrades an order of magnitude slower than RANSAS, while still maintaining the same 100% success rate.

In our second experiment we study the behavior of our algorithms as a function of $|\Omega_k|$, with r = 5, p = $^1/_2$, $\sigma = 0$, and K = 2 fixed. Figures 1 (b) and (b') show that as $|\Omega_k|$ grows, the performance of all methods improves: $\ell_1$ and GREEDYS achieve higher success rates, while RANSAS and K-SPLITS increase their speed and maintain the same 100% success rate.

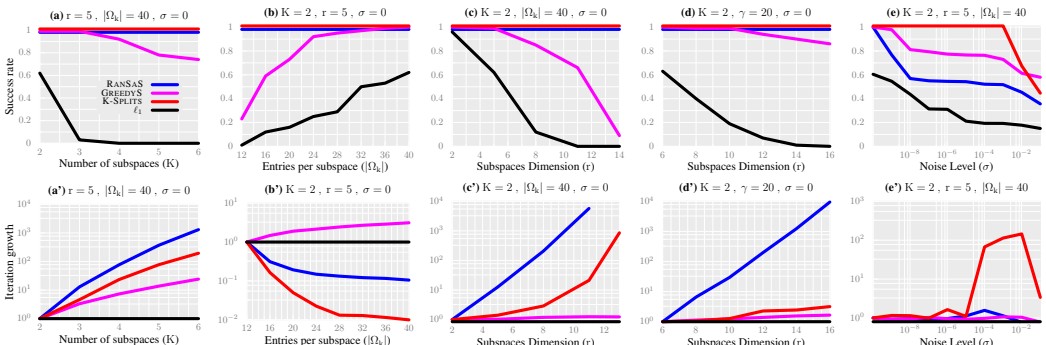

Figure 1: Average success rate and iteration growth over 100 trials, as a function of the number of subspaces K, the number of entries per subspace $|\Omega_k|$ (which in turn is a proxy of the ambient dimension d), the subspaces dimensions r, the gap $\gamma := |\Omega_k| - (r+1)$, and the noise level $\sigma$.

In our third experiment we study performance as a function of r, with $p = 1/2$, $\sigma = 0$, and $K = 2$ fixed. Recall from Lemma 1 that as r increases, so does the number of entries per subspace required for identifiability $(r+1)$. Hence to allow a wide range of r, we fixed $|\Omega_k| = 40$. Figures 1 (c) and (c'), show that (conversely to our previous experiment) the performance of all methods declines as r grows: $\ell_1$ and GREEDYS achieve lower success rates, while RANSAS and K-SPLITS increase their time (though they maintain the same 100% success rate).

Our next two experiments are consistent with the intuition that cases with fewer data (small $|\Omega_k|$) relative to the problem's complexity (the dimension r) are harder. We verify this in our next experiments, where we study performance as a function of r, fixing $p = 1/2$, $\sigma = 0$, and $K = 2$ and the gap $\gamma := p(|\Omega_k| - (r+1)) = 20$. This setting represents the case where we always have a few more (20) entries per subspace than the strictly required $(r+1)$ for splitting. This is arguably a better experiment to test the effect of the subspaces dimensions isolated from the sampling rate. Figures 1 (d) and (d') now show that r slightly affects GREEDYS and K-SPLITS(GREEDYS in terms of accuracy, and K-SPLITS in terms of time), but dramatically affects RANSAS, causing an increase of iterations four orders of magnitude higher than the rest. This is consistent with our discussion in Section 5.1, showing that RANSAC will always succeed given enough time, which grows exponentially in r. Interestingly, K-SPLITS achieves the same 100% performance, but orders of magnitude faster, suggesting that K-SPLITS is in general the most promising SS algorithm.

In our last experiment we study the accuracy of our algorithms as a function of the noise level $\sigma$, with $r = 5$, $p = 1/2$, $|\Omega_k| = 40$, and $K = 2$ fixed. Consistent with Corollary 1, Figure 1 (e) shows that the performance of our algorithms decays nicely with noise, with K-SPLITS allowing up to $\sigma = 10^{-3}$. The price, however, can be seen in execution time (Figure 1 (e')), which grows up to the point where algorithms start to decay, in which case their execution time also starts decreasing. We point out that the fraction of missing data and outliers only affect results and performance in the sense that they reduce the number of available observed entries per subspace, so their effects are also captured in the experiments on $|\Omega_k|$. For example, a case where there are $|\Omega_k| = 40$ samples per subspace, but only half of the entries are observed $(p = 1/2)$ is equivalent to observing $|\Omega_k| = 20$ samples per subspace with no missing data $(p = 1)$. Appendix B shows additional results with $K = 4$, where our methods show even more dramatic improvements.

## BROADER IMPACT

This paper introduces a new dimensionality reduction model, and a series of methods to infer such model, with broad applicability ranging from metagenomics to recommender systems. From a pragmatic standpoint, practitioners can use this research to better predict drug-target interactions, analyze the composition and dynamics soil or human microbiomes, and more. Ultimately, this can impact the well-being of society at many levels, improving drugs, medical diagnoses, treatments, agricultural sustainability, etc. From a learning perspective, we hope this work spurs discussions and insights, and motivates the data science community to explore new directions that stem from this initial work (for example near-optimal initialization strategies on our K-SPLITS algorithm, similar to K-means++), and that leads to better methods and understanding of subspace splitting.

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

# A    PROOFS

In this section we present the proofs of all statements. We begin with the proof of Lemmas 1 and 2, which are required for the proof of Theorem 1.

*Proof.* (Lemma 1) Since $\mathbb{U}^k$ is in general position, if $|\Omega| \leq \dim(\mathbb{U}^k)$, then $\mathbb{U}_\Omega^k = \mathbb{R}^{|\Omega|}$. In other words, $\mathbb{U}_\Omega^k$ is the entire $|\Omega|$-dimensional space. $\square$

*Proof.* (Lemma 2) If $\Omega \subset \Omega_k$ then $\mathbf{x}_\Omega$ trivially lies in $\mathbb{U}_\Omega^k$ by **A2**. To see the reverse implication, let $r := \dim(\mathbb{U}^k)$. We know that $\mathbf{x}_\Omega \in \mathbb{U}_\Omega^k$ if and only if there exists a coefficient vector $\boldsymbol{\theta} \in \mathbb{R}^r$ such that

$$\mathbf{x}_\Omega \; = \; \mathbf{U}_\Omega^k \boldsymbol{\theta}, \tag{1}$$

where $\mathbf{U}^k \in \mathbb{R}^{d \times r}$ denotes a basis of $\mathbb{U}^k$. Since $|\Omega| = r + 1$ by assumption, equation 1 defines a system of $r + 1$ equations (the rows of $\mathbf{x}_\Omega$) and $r$ unknowns (the entries of $\boldsymbol{\theta}$). Assume without loss of generality that $\Omega = \{1, 2, \ldots, r + 1\}$ (otherwise simply permute the rows accordingly), and let $\Omega' = \{1, 2, \ldots, r\}$ be the subset of $\Omega$ with the first $r$ elements. Then we can rewrite equation 1 as

$$\left. \begin{matrix} r \\ \\ 1 \end{matrix} \right\{ \begin{matrix} \\ \\ \{ \end{matrix} \left[ \begin{array}{c} \mathbf{x}_{\Omega'} \\ \hline \mathbf{x}_{r+1} \end{array} \right] = \left[ \begin{array}{c} \mathbf{U}_{\Omega'}^k \\ \hline \mathbf{U}_{r+1}^k \end{array} \right] \boldsymbol{\theta}.$$

Notice that $\mathbf{U}_{\Omega'}^k \in \mathbb{R}^{r \times r}$ is invertible because **A1** guarantees that $\mathbb{U}^k$ is in general position with probability 1. Hence we can use the top block to obtain $\boldsymbol{\theta} = (\mathbf{U}_{\Omega'}^k)^{-1} \mathbf{x}_{\Omega'}$, and we can plug this in the last row to obtain:

$$\mathbf{x}_{r+1} \; = \; \mathbf{U}_{r+1}^k (\mathbf{U}_{\Omega'}^k)^{-1} \mathbf{x}_{\Omega'}.$$

Let $k_i$ indicate the subspace corresponding to the $i^{\text{th}}$ entry of $\mathbf{x}$. Then by **A2** there exist coefficients $\boldsymbol{\theta}^1, \ldots, \boldsymbol{\theta}^K$ such that $\mathbf{x}_i = \mathbf{U}_i^{k_i} \boldsymbol{\theta}^{k_i}$. In light of this we can rewrite the last equation as

$$\mathbf{U}_{r+1}^{k_{r+1}} \boldsymbol{\theta}^{k_{r+1}} \; = \; \mathbf{U}_{r+1}^k (\mathbf{U}_{\Omega'}^k)^{-1} \begin{bmatrix} \mathbf{U}_1^{k_1} \boldsymbol{\theta}^{k_1} \\ \mathbf{U}_2^{k_2} \boldsymbol{\theta}^{k_2} \\ \vdots \\ \mathbf{U}_r^{k_r} \boldsymbol{\theta}^{k_r} \end{bmatrix}. \tag{2}$$

At this point equation 2 has no variables left (all $\mathbf{U}^k$'s and $\boldsymbol{\theta}^k$'s are fixed), which means that equation 2 is a consistency condition. Notice that if $\Omega \subset \Omega_k$, then $k_i = k$, and equation 2 transforms into

$$\mathbf{U}_{r+1}^k \boldsymbol{\theta}^k = \mathbf{U}_{r+1}^k (\mathbf{U}_{\Omega'}^k)^{-1} \mathbf{U}_{\Omega'}^k \boldsymbol{\theta}^k,$$

which further simplifies to $\mathbf{U}_{r+1}^k \boldsymbol{\theta}^k = \mathbf{U}_{r+1}^k \boldsymbol{\theta}^k$. In other words, if $\Omega \subset \Omega_k$, the consistency condition equation 2 is met. On the other hand, if $\Omega \not\subset \Omega_k$, equation 2 does not simplify any further. Recall from **A1** that $\mathbb{U}^1, \ldots, \mathbb{U}^K$ are drawn independently. This implies that the entries in $\mathbf{U}^1, \ldots, \mathbf{U}^K$ keep no relation with one another. Similarly, from **A2** we know that $\mathbf{x}_{\Omega_k}$ is drawn independently. Equivalently, $\boldsymbol{\theta}^1, \ldots, \boldsymbol{\theta}^K$ keep no relation with one another. Intuitively, we can think of the entries in $\mathbf{U}^k$'s and $\boldsymbol{\theta}^k$'s as independent random numbers drawn from a distribution with full measure. We thus conclude that if $\Omega \not\subset \Omega_k$, with probability 1 the consistency condition equation 2 would not hold.

To summarize, we know that $\mathbf{x}_\Omega \in \mathbb{U}_\Omega^k$ if and only if there is a vector $\boldsymbol{\theta}$ that satisfies equation 1. We showed that this will be the case if and only if equation 2 holds, which in turn will happen if and only if $\Omega \subset \Omega_k$, as claimed. $\square$

Notice that if $\Omega$ has $\dim(\mathbb{U}^k) + \ell$ entries, equation 1 becomes a system with $\dim(\mathbb{U}^k) + \ell$ equations, which result in $\ell$ consistency conditions similar to equation 2. By the same arguments as in the proof of Lemma 2, $\mathbf{x}_\Omega$ will agree with $\mathbb{U}_\Omega^k$ if and only if these consistency conditions hold, which will happen if and only if $\Omega \subset \Omega_k$. We thus obtain the following corollary

**Corollary 2.** *Suppose A1 and A2 hold. Let $\Omega$ be an arbitrary subset of $[d]$ with more than $\dim(\mathbb{U}^k)$ elements. Then $\mathbf{x}_\Omega \in \mathbb{U}_\Omega^k$ if and only if $\Omega \subset \Omega_k$.*

With this, we are ready to give a proof of Theorem 1

*Proof.* (Theorem 1) Let $r_k := \dim(\mathbb{U}^k)$. To identify $\Omega_1, \ldots, \Omega_k$ we will exhaustively search for combinations $\Omega$ of $r_k + 1$ entries in $\mathbf{x}$ that match with $\mathbb{U}^k$. By Lemma 2 we know that $\mathbf{x}_\Omega$ will match with $\mathbb{U}^k$ if and only if $\Omega \subset \Omega_k$. By assumption there is at least one k for which $|\Omega_k| > r_k$, so we know that at some point we will find such a subset $\Omega$. Once we do, we can recover $\boldsymbol{\theta}^k = (\mathbf{U}_{\Omega'}^k)^{-1} \mathbf{x}_{\Omega'}$, where $\Omega'$ can be any subset of $\Omega$ with exactly $r_k$ entries (recall that **A1** guarantees that $\mathbb{U}_{\Omega'}^k$ is invertible). Finally, we can compute $\mathbf{x}^k := \mathbf{U}^k \boldsymbol{\theta}^k$, and recover $\Omega_k$ by inspection (the set of zero entries in $\mathbf{x}^k - \mathbf{x}$), where Corollary 2 guarantees that there will be no additional false matching entries. Repeating this procedure for every k we can recover $\Omega_1, \ldots, \Omega_K$, as claimed.

To obtain the converse, suppose that $r_k$ or fewer entries of $\mathbf{x}$ correspond to $\mathbb{U}^k$ for some k. By Lemma 1, any subset $\Omega'$ of $\Omega_k \cup \Omega_0$ with $|\Omega'| \leq r_k$ satisfies $\mathbf{x}_{\Omega'} \in \mathbb{U}_{\Omega'}^k$. Hence we can split the entries of $\mathbf{x}$ corresponding to $\Omega_k$ and $\Omega_0$ in a non-unique manner, and still have them agree with $\mathbb{U}^k$, which makes $\Omega_k$ unidentifiable. □

We now use the same strategy to prove Corollary 1, which extend these ideas to noisy settings.

*Proof.* (Corollary 1) Let $\mathbf{P}_{\Omega'}^k$ denote the projection operator onto $\mathbb{U}_{\Omega'}^k$. Under the assumptions of the corollary, if $\Omega' \not\subset \Omega_k$, then with high probability (decreasing in $C$ and increasing in $c$), $\|\mathbf{x}_{\Omega'} - \mathbf{P}_{\Omega'}^k \mathbf{x}_{\Omega'}\|^2 > |\Omega'| \sigma^2$ (see Theorem 1 in [17]). In other words, the residual of projecting $\mathbf{x}_{\Omega'}$ onto $\mathbb{U}_{\Omega'}^k$ will be too large if the entries of $\Omega'$ do not come from the $k^{th}$ subspace. Hence, instead of searching for a set $\Omega'$ where $\mathbf{x}_{\Omega'}$ perfectly fits in $\mathbb{U}_{\Omega'}^k$ (as in the proof of Theorem 1), we can search for subsets $\Omega' \subset \Omega_k$ where the residual is within the noise level, i.e., $\|\mathbf{x}_{\Omega'} - \mathbf{P}_{\Omega'}^k \mathbf{x}_{\Omega'}\|^2 \leq |\Omega| \sigma^2$. The rest of the proof follows by the same arguments as the proof of Theorem 1. □

## B ADDITIONAL EXPERIMENTS

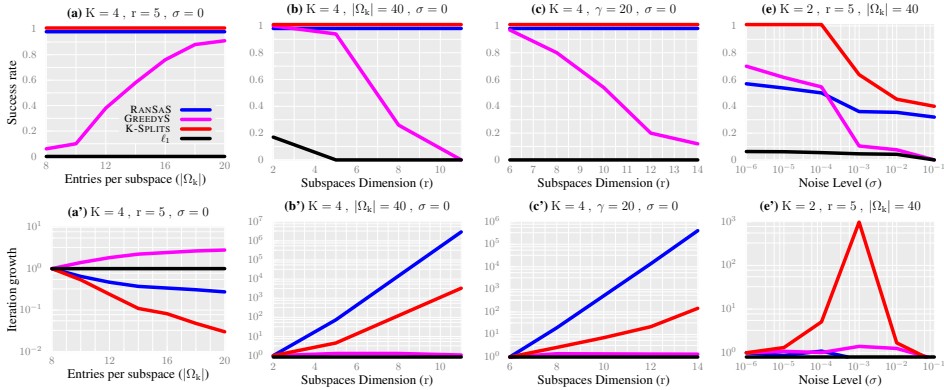

Figure 2: Complementary results to Figure 1 with $K = 4$. Average success rate and iteration growth over 100 trials, as a function of the number of entries per subspace $|\Omega_k|$ (which in turn is a proxy of the ambient dimension d), the subspaces dimensions r, the gap $\gamma := |\Omega_k| - (r + 1)$, and the noise level $\sigma$.

