# OpenReview forum: "Mixed-Features Vectors and Subspace Splitting"
_ICLR.cc/2021/Conference — ICLR 2021 Poster_

### Official Review · AnonReviewer2 · 2020-10-28
**Anonymous review of subspace splitting paper**

**Rating:** 6
**Confidence:** 4

**Review:**

Summary: The paper introduces the problem of subspace spitting, in which an observed mixed-features vector is to be partitioned such that the identified partitions match with given subspaces. The main results of the paper lie in deriving sufficient and necessary conditions for identifiability of these partitions when the subspaces and the entries of the features are randomly positioned in the ambient dimension and the subspaces, respectively. The conditions simply require that there are more entries associated with each subspace than the dimension of the subspace. The paper also presents algorithms to perform the splitting.

Strengths:
- The problem statement is novel. I did not see previous formulations of this problem.
- The paper is generally well-written.
- The paper has a good balance of theory and algorithm development.

Weaknesses:
- The experimental results section is rather weak. It would be good to include some realistic examples from some concrete applications.
- While the paper has a dedicated section on motivating applications, they are not that convincing. The metagenomics application is more plausible, but I do not think this model applies well to recommender systems. The examples provided seem to be included to justify the proposed model. Perhaps there are meaningful relevant applications, but for the current version the problem setup is not sufficiently justified and seems somewhat contrived.
- The assumptions about the subspaces need to be more explicit, especially when discussing the algorithms. For example, the random sampling algorithm seems to require full knowledge of the dimensions of these subspaces which may be impractical.
- I have not fully verified this argument, but I am under the impression that the result of the main theorem is trivial. Isn't it obvious that the span of the restriction of a subspace to a given partition of size m, the whole R^m? I believe the main result can just follow from this simple observation.
- Reproducibility: the authors did not include code for their developed algorithms in the supplementary material.

Impact: Provided that the problem setup and model are better justified, I believe this work could open up new research questions in machine learning and data analysis, including (mixture) variants of well-studied problems on matrix completion and robust learning.

Overall I like the paper, especially that the problem setup itself seems novel. However, the model is not sufficiently justified, the assumptions are somewhat questionable, and the experimental section is lacking. As such I would rate this as a marginal acceptance.

---

> ### Author Response · Authors · 2020-11-25
> **Response to Reviewer 2**
>
> We thank the reviewer for their comments.
>
> This paper introduces the mixed features vectors model, and the subspace splitting problem. As such, our main focus was to explore its feasibility, motivate its applicability, and start developing fundamental theory. Now that we have established these points, our future work will focus on detailed and thorough experiments on real-data, which will require special considerations that are out of the scope (and length) of this paper.
>
> We will be happy to make more explicit the assumptions of each algorithm. To clarify, rather than knowledge of the exact dimension of the subspace, an upper bound will suffice for the random sampling algorithm (the price that one pays for the bound gap is an increased computational complexity, as implied by Theorem 2). We are happy to include a remark about this.
>
> As for the triviality of our main Theorem, you are partially correct. The proof of such Theorem requires Lemmas 1 and 2. Lemma 1 is a sort of *if* condition, and Lemma 2 if a sort of *only if* condition. As is often the case, one direction is easy/trivial, and the other one isn't. Lemma 1 is the easy direction, and indeed follows as direct consequence of your argument [that the span of the restriction of a subspace of dimension $\geq m$ in general position, to a given partition of size $m$, will be the whole $\mathbb{R}^m$]. Lemma 2, however, is a little more subtle. It requires to show that *only* a *true* combination of $r+1$ coordinates will fit into one of the *true* $r$-dimensional subspaces. Showing that the event that a *false* combination of $r+1$ coordinates fits into a *true* $r$-dimensional subspace has measure zero requires some care, and is non-trivial. Having said that, one can argue that even Lemma 1 is non-trivial; for instance, Reviewer 4 who would like to see more details about this point.
>
> We apologize for the code. We decided to avoid any pointers to preserve anonymity, as required by the conference guidelines. We are happy to provide a link and references to our github and websites upon acceptance.

---

### Official Review · AnonReviewer4 · 2020-10-28
**interesting new setting but with limitations**

**Rating:** 6
**Confidence:** 3

**Review:**

This paper introduced a new setting in which variables in feature vector are divided into different groups and in each group, variables are generated by sampling within linear subspace. They characterized the sufficient and necessary condition for the system to be identifiable. Three algorithms are provided to cluster the variables into their generative subspace. They also provide motivating applications for this new setting in metagenomics, recommender systems, and robust learning. I think the new setting is potentially interesting and the proposed algorithms are good. However, compared with, for example, subspace clustering, the generative subspace {U_i} is known to the users a prior and the goal is to classify variables of a single data point, which might be rarely the case in practice.

Comments:

My main concern is on the proof of Theorem 1, which seems a little bit sloppy and not convincing enough to me. For example, on page 14, in the mid of Eq.(1) and Eq.(2), the authors argue U_\Omega^k is invertible as subspace U^k is in so-called general space. Please elaborate on that. I am confused as if we set U_\Omega^k any full rank r-by-r matrix and let U_{r+1}^k be a copy of any row in U_\Omega^k. And, do a row swap between r+1 and any row in [r]. Then, the new r-by-r U_\Omega^k cannot be invertible as it is not full rank in row. It is not clear to me the set of this type of subspace a zero-measure set on the uniform distribution of (d,r)-Grassmannian. Please consider rewriting the proof in a more rigorous way.

---

> ### Author Response · Authors · 2020-11-25
> **Response to Reviewer 4**
>
> We thank the reviewer for their comments.
>
> The key observation here is that under A1, since each subspace is in general position, the probability that two rows of its basis are linearly dependent by mere chance, is zero. One can certainly construct a "degenerate" subspace that has linearly dependent rows, like the one you suggest. However, if we draw one subspace uniformly at random from the Grassmannian (as specified in A1), then the probability of drawing any such subspace is zero. This is because the set of all $r \times r$ rank-deficient matrices is described by a polynomial manifold that has measure zero. This manifold is called the determinantal manifold, precisely because an $r \times r$ matrix will be linearly dependent only if its determinant is equal to zero, which is a polynomial constraint of measure zero. Equivalently, under A1, any set $\Omega$ with exactly $r$ entries will produce, with probability 1, a generic matrix $U_{\Omega}$ of size $r \times r$ with linearly independent columns, which implies it must be invertible.
>
> We are happy to include this discussion as part of the proof of Theorem 1 in the final version.

---

### Official Review · AnonReviewer1 · 2020-11-10
**Mixed-Features Vectors and Subspace Splitting**

**Rating:** 6
**Confidence:** 3

**Review:**

The authors propose a method to perform subspace splitting. That is, the task of clustering the entries of an input vector into sets of coherent subspaces. The contribution of the work is two-fold: (1) the theoretical characterization of the problem, and its well-posedness, and (2) the presentation of three algorithms for tackling the problem of subspace splitting. Quantitative analysis of the performance of the three algorithms is provided by means of synthetic experiments.

The paper is well written, with sound mathematical formulation. The contributions proposed by the authors seem to have enough novelty and relevance for the community, and both theoretical and practical contributions are thoroughly motivated and discussed.

Major concerns:

* I find the structure and content of the paper somehow unbalanced. A fairly large portion is dedicated to motivational applications, which feels like an extension of the related work section, but are not experimentally explored. In Section 5 most discussion is devoted to approaches with clear drawbacks (RanSaS, GreedyS), while the final proposal (K-splits) is hardly discussed.

* While it is true that a large contribution of the work is strictly theoretical, the practical aspect of the paper could be further backed by experimental validation. The experiments section show limited results regarding the choice of the number of clusters (only 2 or 4). The performance of RanSaS and K-Splits is saturated in all noise-free experiments which makes them hard to compare or identify failure cases.

* Given the attention paid in the paper to motivating applications (a full section) I miss a section in the Experiments where the proposed approach is validated with real data from any of the mentioned applications. I feel this would tremendously increase the value of the work and would legitimize the claim of practical importance that the authors make in introduction and conclusions.

Minor concerns:

* At the end of the first paragraph of related work, the authors claim that random sampling is not applicable to subspace splitting, with no further explanation of why this is the case. At the same time the algorithm presented (RansaS) is based on random sampling which seems contradictory.
* As I mentioned earlier, I miss more discussion regarding the K-splits approach. Did the authors find any clear drawbacks besides initialization? What about the K-means assumption of isotropic clusters?
* The choice of just l1 as a baseline for comparison is a bit arbitrary. I wonder why the other mentioned approaches (mixed-integer programming or random sampling) were not added to the evaluation.
* The paper could use further review to avoid typos and missing references.
* The paper doesn’t follow the citation convention: authors’ last names and year.

---

> ### Author Response · Authors · 2020-11-25
> **Response to Reviewer 1**
>
> We thank the reviewer for their comments.
>
> This paper introduces the mixed features vectors model, and the subspace splitting problem. As such, our main focus was to explore its feasibility, motivate its applicability, and start developing fundamental theory. Now that we have established these points, our future work will focus on detailed and thorough experiments on real-data, which will require special considerations that are out of the scope (and length) of this paper.
>
> We also understand the concern regarding the saturated performance for K-splits and RanSaS. This is due to a shared property by both systems: they always reach the solution given enough time. This is also the reason why as the complexity of the problem increases, they are the only ones that produce a solution. The counterpart for K-splits and RanSaS is the time it takes for both to reach the solution. For this reason it was decided to add the bottom row of graphs: to show the increase in time it takes for the algorithms to reach a solution.
>
> We couldn't use mixed integer programs as baselines due to their computational limitations to handle large number of variables, in light of their exponential complexity. We indeed study random sampling; it is in fact one of our main methods; see Section 5.1.
>
> The choice of small values of K was made because (as shown in our experiments), existing baselines cannot handle much more than that. However, we indeed compare against larger values of K. See for example the bottom-left plot in Figure 1.
>
> Regarding the analysis of our more promising K-splits algorithm, we point out that it is inspired in K-means, but it has fundamental differences that require special care. For example, the assignment is done in a cyclical manner. This eliminates the isotropic assumption of K-means. Instead it is assumed that each cluster has the same number of elements, which, as mentioned in the paper, is the most complicated case to solve. Even if the assumption is not fulfilled, it is enough for the algorithm to detect one correct cluster in order to be able to exclude it from future iterations, so this assumption does not affect the effectiveness of the algorithm. This is one example of the subtleties of K-splits that require a careful analysis that is out of the scope of this paper, but that will certainly be the focus of our future work.
>
> We will be happy to clarify all these points in the final version.

---

### Author Response · Authors · 2020-11-25
**Response**

We thank the reviewers for their comments. We address specific concerns below.

---

### Decision · Program_Chairs · 2021-01-07
**Final Decision**

**Decision:**

Accept (Poster)

**Comment:**

The paper considers a new linear-algebraic problem motivated by applications such as metagenomics which requires the algorithm to partition the coordinates of a long noisy vector according to a few known subspaces. A number of theoretical questions were asked (e.g., identifiability;  efficient algorithms and their error bounds; etc).

The reviewers generally liked the paper for what it does. Specific suggestions were raised by the reviewers, including how the paper went into length about the motivating applications but did not end up evaluating the proposed algorithms on any motivating applications; and that the main theoretical results were not technically challenging / nor surprising (although the authors provided a fair justification in their rebuttal).

The AC finds the paper an outlier in terms of the topics among papers typically received by ICLR, but liked the paper precisely because it is different.  The authors are encouraged to discuss the connections of the specific problem to the context of representation learning and machine learning in general.

Overall, I believe the paper is a solid borderline accept.